# The Breast Cancer Protein Co-Expression Landscape

**DOI:** 10.3390/cancers14122957

**Published:** 2022-06-15

**Authors:** Martín Ruhle, Jesús Espinal-Enríquez, Enrique Hernández-Lemus

**Affiliations:** 1Computational Genomics Division, National Institute of Genomic Medicine, Mexico City 14610, Mexico; martinruhle@gmail.com (M.R.); jespinal@inmegen.gob.mx (J.E.-E.); 2Center for Complexity Sciences, Universidad Nacional Autonoma de Mexico, Mexico City 04510, Mexico

**Keywords:** breast cancer, proteomics, co-expression networks, protein–protein interactions

## Abstract

**Simple Summary:**

Proteins are among the most fundamental building blocks and molecular players behind the functions of cells and tissues. Their abundance and interaction patterns shape, to a large extent, what happens at the cellular and organ levels. This is also true regarding tumor tissues. In this work, we explored the patterns of abundance and co-occurrence of a large number of proteins in breast cancer cells and their healthy counterparts. We discovered the main differences and tried to see whether those differences may be associated with relevant aspects of the biology of these tumors. Our final goal is to provide information to empower cancer clinicians and pharmacologists to develop better diagnostic, prognostic, and therapeutic tools.

**Abstract:**

Breast cancer is a complex phenotype (or better yet, several complex phenotypes) characterized by the interplay of a large number of cellular and biomolecular entities. Biological networks have been successfully used to capture some of the heterogeneity of intricate pathophenotypes, including cancer. Gene coexpression networks, in particular, have been used to study large-scale regulatory patterns. Ultimately, biological processes are carried out by proteins and their complexes. However, to date, most of the tumor profiling research has focused on the genomic and transcriptomic information. Here, we tried to expand this profiling through the analysis of open proteomic data via mutual information co-expression networks’ analysis. We could observe that there are distinctive biological processes associated with communities of these networks and how some transcriptional co-expression phenomena are lost at the protein level. These kinds of data and network analyses are a broad resource to explore cellular behavior and cancer research.

## 1. Introduction

Breast cancer is the first cause of death in young women worldwide [1]. In spite of being one of the most-studied diseases, a complete, integrative understanding of the mechanisms underlying the cancer phenotype is still missing. Within this challenge, high-throughput omic technologies have provided us unprecedented tools to study cancer at a deeper level. In particular, the study of protein expression, due to the advances in mass spectrometry, has gained greater relevance in the last few years and is promising a new era of proteomics-driven precision medicine [2]. As in genomic sequencing, these advances in mass spectrometry in recent years, and the open data practice, have provided us with a great amount of information at the protein level [3], and for this reason, computational proteomic analysis takes priority and relevance to interpret tumor biology in the context of proteomic data.

A common procedure to evaluate a given phenotype is the analysis of gene expression profiles. However, the information of the interactions appearing between genomic species is missing with this technique. To overcome this lack of interaction information, several approaches have been developed, in particular those based on measures of the statistical dependency of the gene expression profiles. This way, co-expression networks may point out to gene sets that are functionally related or controlled by the same regulatory program [4].

The Pearson correlation coefficient is generally used to measure the dependency between variables. The main limitation with the use of the Pearson coefficient is that it only detects the linear relationships between variables, and in the expression of biological components, such as genes or proteins, it is common that they do not follow a linear dependency [5,6]. To capture all the possible relations between the expression of the biological components, a better approximation can be their mutual information profiles. The mutual information of two random variables measures the reduction in uncertainty of one variable *x* due to the knowledge of the value of another variable *y*. In this way, we use the mutual information between the protein levels or gene expression data to infer co-expression networks and analyze the relationship between the different biological components involved in the tumor molecular physiology.

Despite high expectations, proteomics-based network research has experienced only moderate growth [7]. Therefore, integrating and complementing genomic analysis with the proteomic approach and the application of networks will result in being of relevance to improve our understanding of the underlying biology, as well as the classification and diagnosis of breast cancer. For these reasons, here, we studied the breast cancer proteome profile with a co-expression networks approach.

By analyzing the protein co-expression network, we are able to integrate the protein co-expression landscape. This approach provides a robust set of the protein interactome in terms of statistical dependency between those variables. The protein coexpression network (PCN) can be compared with other data-driven inferred networks, in order to assess the similitude or differences between distinct levels of description. The PCN community structure can also reveal functional features of the protein interactome and be complementary to the differential expression landscape. With this approach, we are able to investigate whether functional communities in the breast cancer PCN are similar in terms of functions.

## 2. Methods

Figure 1 shows the workflow followed in this work. It can be summarized in 4 main stages: (1) data acquisition, quality control, and data curation, (2) differential protein abundance and pathway analysis, (3) inference and analysis of protein co-expression networks, and (4) network modularity analysis and protein interaction networks. These stages will be discussed in the following subsections.

### 2.1. Stage 1: Data Acquisition, Quality Control, and Data Curation

As shown in Figure 1, our pipeline starts with data acquisition from the primary databases and preliminary curation and quality control (QC). Since proteomic data obtained from mass spectrometry experiments are quite noisy and present some measurement biases due to the associated experimental techniques, a QC stage prior to any analysis is both customary and recommended [8,9].

A proteogenomic expression matrix, generated from the proteome mass spectra of 105 breast cancer samples, along with 3 healthy tissue samples and 3 replicates of tumor samples, was acquired from the Clinical Proteomic Tumor Analysis Consortium (CPTAC), which now is part of Proteomic Data Commons (PDC) project, which in turn acquired the samples from the Genomic Data Commons (GDC). There are 10625 protein-derived “*genes*” reported (as these *genes* are actually inferred from the protein abundance, henceforth, in this work, we will call them *pGenes*), and the 4 subtypes of breast cancer (Luminal A, Luminal B, basal-like, HER2-enriched) are represented. The methodology used for the generation of mass spectra is specified in the following link: https://pdc.esacinc.com/pdc/browse(studySummary:study-summary/TCGA_Breast_Cancer_Proteome) (accessed on 26 April 2022). A quality control analysis of the proteogenomic matrix was performed using R. The specific libraries used included ggplot2 [10], dplyr [11], and tidyverse [12].

QC proteomic analysis proceeded as follows: A proteogenomic expression matrix generated as in the reference above was downloaded from the CPTAC/GDC site. Since this matrix was incomplete (many pGenes were not available for all the samples), we filtered out the matrix. All pGenes with missing information in more than 60 samples were discarded. This accounts for less than 10% of the total molecules measured removed. Furthermore, mining metadata from the CPTAC database, we found that 28 samples presented significant protein decay and may induce background noise or biases in our analysis. Those samples were removed from our study. Finally, in order to perform contrasts with the healthy tissue samples, pGenes that were not measured in the 3 healthy samples were also removed from our analysis.

### 2.2. Stage 2: Differential Protein Abundance and Pathway Analysis

Once protein expression levels and quantitative measurements for *pGenes* have been reliably obtained, we may proceed to stage 2 of our study, which consists of differential expression analysis (in this case of pGenes, which, as we have said, place protein expression levels in the context of the genes associated with the measured polypeptides) and functional enrichment to advance biological hypotheses regarding the different phenotypes.

The differential expression analysis consists of determining if there is a significant difference in the expression of the proteins of the tumor tissue samples compared to the samples of healthy tissue. This analysis is based on an adjustment of linear models using the R-Bioconductor package, linear models for microarrays, and RNA-seq limma [13], and the plotting was performed in Python, using the specific libraries seaborn [14], matplotlib [15], and pandas [16].

Gene set enrichment analysis (also called over-representation analysis) allows determining which cell pathways or processes are significantly deregulated. Here, we used the online tool WebGestalt (WEB-based GEne SeT AnaLysis Toolkit), which, through hypergeometric tests, determines the probability that the unknown gene set of interest is associated with a particular cell pathway or biological process. This tool uses the Gene Ontology (GO) and Kyoto’s Encyclopedia of Genes and Genomes (KEGG) databases and was used to analyze both differentially expressed *pGenes* and *pGenes* belonging to the different communities of the generated networks, as detailed later.

The parameters used for the biological processes’ enrichment analysis were:Type of process: biological process non-redundant;Number of genes per category: 5–2000;Fit test: Benjamini and Hochberg (BH);Multiple testing correction: false discovery rate (FDR);Reference gene list: genome protein-coding.

### 2.3. Stage 3: Inference and Analysis of Protein Co-Expression Networks

We used mutual information as a measure for co-expression since it has the ability to capture any statistical dependence between the expression of different genes or proteins. This characteristic is advantageous since there are many relationships between genes or proteins that do not follow a linear statistical dependence.

ARACNe and Cytoscape were used, respectively, to generate, visualize, and analyze mutual information coexpression networks. The Algorithm for the Reconstruction of Accurate Cellular Networks, Adaptive Partitioning strategy (ARACNe-AP) [17] was used for the calculation of the values of mutual information between the different pairs of *pGenes*, based on the expression values of the proteogenomic matrix. The second tool Cytoscape, refs. [18,19] was used for network visualization and subsequent topological analysis.

The networks generated were based on the tumor data alone, since the low number of control samples (three) was not enough to infer good-quality co-expression networks. To compare the topology of the protein co-expression network with the gene co-expression network, we used the network generated by Espinal-Enriquez et al. [20] since they used the same (transcriptome) samples and methods we used in this work.

We will detail below the different parameters used to reach the final result of the network.

The ARACNe algorithm (available at https://sourceforge.net/projects/aracne-ap/files/ (accessed on 26 April 2022)) was followed to infer the network. The *p*-value chosen to generate the cut-off point of the interactions was $p<10^{-8}$.

Within Cytoscape, from public databases (BioMart), tables containing the information of which chromosome each gene belongs to were downloaded (the human genome GRCh38.p13), to analyze whether there is an expression grouping correlated with the chromosomal location.

### 2.4. Stage 4: Network Modularity Analysis and Protein Interaction Networks

Another tool we used from the Cytoscape network analysis suite was cluster making, community cluster analyzer Glay, an application that combines different community detection algorithms [21].

The Search Tool for the Retrieval of Interacting Genes/Proteins (STRING) database was used to generate the molecular protein interaction networks. The pGene sets belonging to the communities, generated by Glay, from the mutual information networks were selected to evaluate the presence of interactions reported in validated protein–protein binding experiments between these groups of proteins. This is so, since protein co-expression may in some cases obey features related to biological function that may in turn be captured by the formation of protein complexes.

As explained in stage 2 we performed enrichment analysis of the pGene network communities detected to determine the possible biological functions of each community, looking for the functional basis of coexpression phenomena.

## 3. Results

### 3.1. The Protein Co-Expression Network Shows a Scale-Free Topology

We found that the connectivity distribution of the breast cancer PCN has a power-law-like (scale-free) behavior that is known to be characteristic of complex networks, in general, and of biological networks, in particular [22,23]. Indeed, this long-tail behavior of the degree distribution has been recently associated with the response of the underlying systems to perturbations [24].

The log–log degree distribution plot is a well-known parameter to determine the topology regime of a given network [25,26], though more rigorous statistical approaches to the goodness of fit have been also developed [27]. As can be observed in Figure 2, the degree distribution plot follows a well-fit power law distribution (α=−1.495,Corr=0.999).

### 3.2. There Is a Hierarchical Organization in the PCN

The top-10,000 highest mutual information (MI) interactions between proteins in the breast cancer network can be observed in Figure 3. There, network nodes are depicted according to the community to which each node belongs. Importantly, communities in the first level of organization are large; however, they can be iteratively organized into subcommunities, such as the case of insets B and C of Figure 3. The pink nodes (community 2), surrounded by an ellipse, can be divided into 25 subcommunities, each one of them with a different number of elements. The same case is observed in the square-delimited green nodes (community 1), where seven subcommunities are found.

It is precisely at the level of subcommunities where we analyzed the functional implications of those groups of proteins and the association with biological functions.

### 3.3. Subcommunities Are Significantly Associated with Specific Biological Processes

One of the most relevant aspects in which community structure takes place in biological networks is precisely the functional aspect, i.e., whether the molecules present in a given community are associated with a particular biological function [28,29]. To assess the latter, we performed over-representation analysis [30] in the subcommunities obtained by the given partition.

In Figure 4, the significance values of the aforementioned subcommunity enrichment analysis are presented. The heat map presents enriched categories (GO: biological process, rows) in all subcommunities (columns). It is worth noticing that in the figure, $p_{val}>10^{-5}$ are depicted in white. Three aspects result remarkably from the figure: (1) the high significance of certain processes ($p_{val}<10^{-50}$), (2) the specificity of enriched communities (only a few and compact red squares), and (3) the similitude between certain communities in terms of similarly enriched processes (upper dendrogram).

### 3.4. Subcommunities with Similar Enrichment Are Part of the Same Community, and They Share Several Interactions

The subcommunities found in the PCN of breast cancer show a significant association with specific biological processes. As mentioned before, certain subcommunities resulted in being associated with similar processes. This is the case of subcommunities 1, 4, and 5 of community 5 (from now on, these subcommunities will be named $SC_{5\_1},SC_{5\_4},SC_{5\_5}$, etc.).

Figure 3 shows the three cases also highlighted in Figure 4. The red links represent interactions between subcommunities with similar processes. For example, in the case of $SC_{5\_1},SC_{5\_4},SC_{5\_5}$, it can be observed that a high number of links exist between $SC_{5\_4}$ and SC5_5. The links of those two subcommunities with $SC_{5\_1}$ are more scarce, but still important. The processes associated with those subcommunities are immune-related processes. For the case of $SC_{21\_2}$ and $SC_{21\_3}$, the associated processes are translation and gene expression. Finally, $SC_{2\_1}$ and $SC_{2\_6}$ share several metabolism-related processes. The subgraph structure of these networks is shown in Figure 5. All enriched categories for all subcommunities are reported in Appendix A. It is worth noticing that the statistically significant enriched functions do not overlap between communities, but the overlap is important between subcommunities, despite said groups being mutually exclusive, i.e., there are no shared nodes (proteins) between subcommunities.

### 3.5. The PCN Is Clustered by Differential Expression Trend

The protein co-expression network used 10,900 *pGenes*, which in turn may result in a network of more than 100 millions of interactions. In order to be able to analyze the network, we decided to conserve the top 0.1%, i.e., the highest 10,000 interactions. The resulting network is observed in Figure 6. The proteins in the network are depicted according to their differential expression values. Red nodes represent overexpressed proteins, while the blue ones correspond to underexpressed molecules. In the network, it is possible to observe that nodes are clustered in an important way according to their differential expression trend: overexpressed protein are together, as well as underexpressed ones. To quantify the trend of differential expression in the PCN, we calculated the *expression assortativity* for each community in the network.

Expression assortativity was calculated as in [31]; in that work, the log2-fold change sign was used to classify gene expression assortativity: either overexpressed or underexpressed.

Appendix A shows the expression assortativity for each subcommunity with more than 10 interactions. In total, 58 subcommunities were used to observe the trend of differential expression. Twenty-three communities have $exp_{ass}<0.5$, while 35 subcommunities have values close to 1. Additionally, Appendix A presents the full proteomic differential expression analysis results.

### 3.6. Protein Co-Expression Is Not Distance-Dependent

There have been previous reports performed by our own group where it was shown how gene co-expression networks (GCNs) have a remarkable tendency to present the large majority of top interactions between genes from the same chromosome, i.e., networks are composed by interactions between genes from the same chromosome [20,32,33]. This phenomenon, called *loss of long-range co-expression*, was firstly observed in non-classified breast cancer [32]. After that seminal work, the GCNs from breast cancer molecular subtypes [34] were also described with the loss of long-range co-expression, but more importantly, intra-chromosome interactions did not occur randomly in the entire section of the chromosome; the majority of top interactions occurred between neighbor genes in terms of the cytoband in which those genes were located. This phenomenon is called gene co-expression distance dependency (GCDD) [35,36].

In order to observe whether the GCDD occurs also in the PCN, we calculated the fraction of intra-chromosome interactions. Surprisingly, only 3073 out of 18,735 are inter-chromosomal interactions; while 15,802 interactions are trans, a result in stark contrast with the one found in gene co-expression networks.

As can be clearly observed from Figure 7, the phenomenon of the loss of long-distance co-expression is not maintained in the PCN. Furthermore, the size of components in the gene co-expression network is substantially lower than that in the giant component of the PCN. However, as previously mentioned, the community structure with functional associations prevails in the PCN.

## 4. Discussion

Protein co-expression networks are emerging as a valuable tool for the study of complex phenotypes, including pathological states such as tumors [37,38,39]. Gene co-expression networks have been quite successful, providing molecular insights into tumor biology, yet for obvious reasons, these are not able to account for post-transcriptional, translational and post-translational phenomena, which may constitute important players in cancer. Protein co-expression networks are indeed a more natural object to study, but it is only recently that the technology to accurately measure protein levels in complex tissues has become precise enough to provide a basis for large-scale quantitative modeling.

In this context, and in connection with breast cancer, we should remark about the recent contribution by Yamada and coworkers [40]. The authors performed a detailed analysis of protein co-expression networks on a rigorously selected set of 10 lesions in five estrogen-receptor-positive/HER2-negative/Ki-67-positive luminal breast tumors. They analyzed a set of 5 high-Ki-67 expression lesions (HOT) and 5 (paired) low-Ki-67-expression lesions (COLD). Proteomic MS/MS quantitative measurement data were modeled using correlation network analysis (WGCNA) followed by pathway enrichment by over-representation to reveal functional clusters. Pathway enrichment results were supplemented with causal network analysis to reveal functions related to ribosome-associated quality-control pathways, heat shock response by oxidative stress and hypoxia, angiogenesis, as well as oxidative phosphorylation. The authors argued that some of these functions may actually lead to the design of therapeutic interventions.

The approach followed in our work is indeed in stark contrast with the one by Yamada et al. The present study considered a much larger set of breast cancer tumors, representing all major breast cancer subtypes. The fact that we studied both a larger and wider dataset allowed us to focus on more general properties of the protein co-expression phenomenon. We are somehow gaining in *field* and losing *focus*. Hence, we will describe a panoramic view of protein coexpression in breast tumors.

Along these lines, the fact that a relevant number of PCN clusters are formed by subunits of the same protein may imply that post-translational effects are, indeed, more determinant for co-expression of those elements than expected [20,33]. Additionally, a functional role of the components of each protein co-expression community or module emerges, in contrast to the distance-dependent phenomenon previously observed in GCNs [35].

The functional character behind the PCN structure is further highlighted in the context of differential expression patterns. The assortativity signatures in protein co-expression is clearer in the functional clusters.

As mentioned in the Results Section, as well as in Figure 6, several subcommunities present high values of assortativity; moreover, subcommunities with high assortativities also present statistically significant enriched categories. Such is the case of subcommunities $SC_{5\_1},SC_{5\_4},SC_{5\_5}$, strongly associated with immune system processes (Figure 4 and Figure 5).

Importantly, for the case of $SC_{21\_2}$ and $SC_{21\_3}$ (Figure 5, community 21), translation and gene expression are highly enriched. This is a particular case since those communities are composed of similar elements: $SC_{21\_2}$ contains 19 out of 21 MRPS pGenes; on the other hand, $SC_{21\_3}$ contains 38 out of 40 MRPL pGenes. Interestingly, $SC_{21\_3}$ has high assortativity values with a clear trend of overexpression. Conversely, MRPS pGenes do not have a clear tendency of overexpression or underexpression.

The hierarchical structure of communities in the PCN is also relevant. The similarities in processes of communities with shared edges also suggest a kind of orchestrated functionality between co-expressed proteins. This is another instance in which the structure can be strongly associated with the function in a biological network [29]. It is important to mention that the inferred communities are mutually exclusive, i.e., there are no shared proteins among communities.

Regarding the observed enriched process, certain subcommunities resulted in being associated with similar processes. This is the case of subcommunities 1, 4, and 5 of community 5, $SC_{5\_1},SC_{5\_4},SC_{5\_5}$. In particular, for $SC_{5\_1}$, a subset composed of the GBP1 to GBP7 genes is indeed part of that subcommunity. These proteins are induced by interferon, signaling molecules from the host cells to regulate immune response [41]. Additionally, HLA-A, B, C, E, F, G, and H appeared together in the same subcommunity. Those sets are a crucial part of the adaptive immune system. Immunoproteasome (IP) genes PSMB8, PSMB9, and PSMB10 are also part of $SC_{5\_1}$, and they have interactions with subcommunities $SC_{5\_4},SC_{5\_5}$. On the other hand, $SC_{5\_5}$ contains families of genes such as RHO GTPase-activating proteins (ARHGAPs), HLA family, or GIMAPs (Proteins of the GTP-binding superfamily and, in particular, of the immuno-associated nucleotide (IAN) subfamily of nucleotide-binding proteins. These genes are located in a cluster at 7q36.1 and are arguably involved in the differentiation of T helper (Th) cells of the Th1 lineage. The related mouse gene is known to be essential for the development of mature B and T lymphocytes [42,43]. Read-through transcription happens between this gene and the downstream GIMAP5 gene).

As can be observed, families of genes in the three subcommunities of community 5 are certainly related and participate in similar, yet complementary processes. Here, we can hypothesize that protein co-expression is more closely related to the functionality than in the genomic location of their coding genes.

For the case of $SC_{21\_2}$ and $SC_{21\_3}$, as shown in Figure 5, several points are important to mention. For instance, $SC_{21\_3}$ is composed of 44 proteins. Only 4 out of those 44 are not part of the MRLP family; in other words, a community of 44 proteins contains 40 large-subunit mitochondrial riboproteins. Processes associated with this subcommunity are translation and biosynthesis, i.e., regulation of gene transcription. For the case of $SC_{21\_2}$, analogously, a short subunit of mitochondrial ribosomal proteins (MRPSs) is the majority in this subcommunity (19 out of 33). Community 21 contains a large amount of small and large subunits of mitochondrial ribosomal proteins. Intriguingly, one subcommunity contains a short subunit and the other one presents those corresponding to the large subunit of mitochondrial riboproteins.

Finally, $SC_{2\_1}$ and $SC_{2\_6}$ share several metabolism-related processes. Those subcommunities are rich in zinc fingers (63). Subcommunity $SC_{2\_1}$ is highly enriched for terms associated with metabolic processes, such as $SC_{2\_6}$. However, the first one also contains SRSF genes, which correspond to serine- and arginine-rich splicing factors. These proteins can either activate or repress splicing [44]. It is worth noticing that the enriched functions are not overlapped between communities, but the overlap is important between subcommunities, despite said groups being mutually exclusive, i.e., there are no shared nodes between subcommunities.

The fact that both subcommunities are part of the same large community, as well as each one of them presents an exclusive group of MRPs comprise a paradigmatic example of how the modularity observed in the hierarchical community structure of the PCN favors the synergy between complementary sets of co-expressing proteins. In other words, the hierarchical level of the community structure allows us to observe complementary, yet separated protein sets to perform a synergistic phenomenon, in this case the formation and activation of mitochondrial ribosomes.

In the resulting protein co-expression networks—based on robust mutual information statistical dependencies—we do not observe the phenomenon observed at the transcription level, where co-expression is strongly associated with chromosomal location in tumor phenotypes [32,34,35,45,46]. This result shows that there are remarkable differences between the cancer regulation programs at the transcription and the translation levels, indicating the importance of multiomics analysis to understand the general biomolecular expression regulatory program. Efforts in this line may include (but are not limited to) the analysis of micro-RNA-gene co-expression analysis [47,48], copy number alteration’s role in the co-expression landscape [31,49], epigenetic regulation [50], among others.

## 5. Conclusions

By analyzing the protein co-expression network communities, we could see that many of the modules generated respond to characteristic cancer processes, which are closely linked to the paradigmatic *Hallmarks of Cancer*, as well as to various well-known protein complexes [51]. Furthermore, protein–protein interactions were enriched in the co-expression network. These results respond to the functionality of proteomic expression, which may be associated with the survival needs of the tumor cells, thus highlighting the usefulness of large-scale analytic approaches to mutual information co-expression network analysis to further our understanding of tumor biology.

This study has intrinsic limitations. One of them is related to the number of samples. Firstly, the number of non-tumor samples was not enough to perform a mutual information analysis. To or knowledge, there is not a set of non-cancer breast tissue samples analyzed with MS or a similar technology yet. Endeavors to collect non-tumor samples to provide that information will be beneficial in several approaches.

Similarly, given that mutual information requires a high number of samples to preserve the statistical significance of the protein co-expression interactions, the network inferred here was constructed with unclassified breast cancer samples; therefore, we were not able to separate into molecular subtypes the cancer samples. Future directions must include molecular subtype classifications.

Finally, this work must be nurtured by different, yet complementary omic approaches, such as genomic analyses, micro-RNA/long non-coding RNA, Chip-seq, Hi-C, methylation, copy number alterations, and even the mutational profile of the same samples. However, the research presented here points to an undeniable relationship between protein co-expression and biological function in breast cancer. The interconnection with the other levels of description will allow understanding in depth the causes and mechanisms behind this disease.

## Figures and Tables

**Figure 1 cancers-14-02957-f001:**
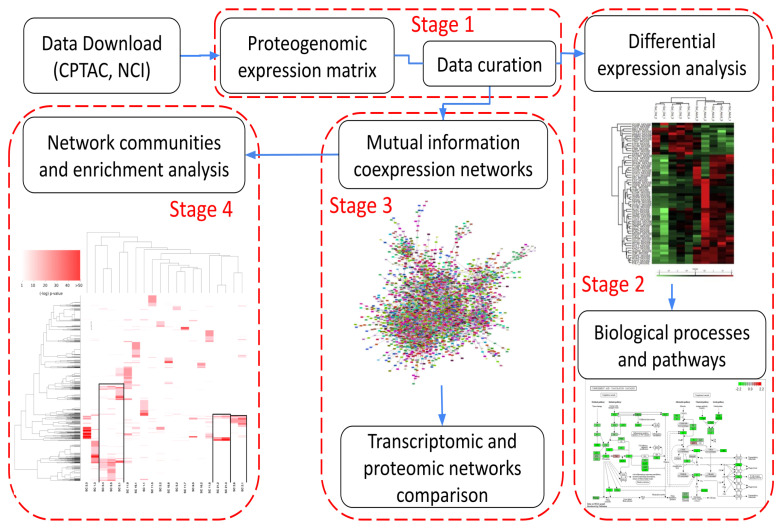
Workflow used in this work.

**Figure 2 cancers-14-02957-f002:**
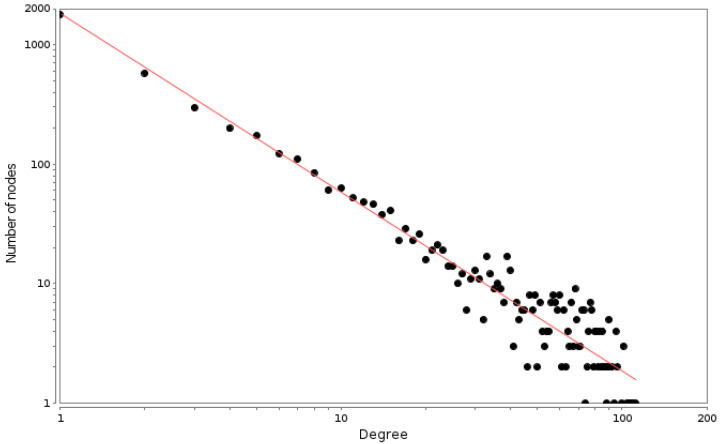
PCN degree distribution plot. Each point represents the number of proteins (Y axis) that have a given number of neighbors (X axis). The red line is the power law fitting with an exponent α=−1.495.

**Figure 3 cancers-14-02957-f003:**
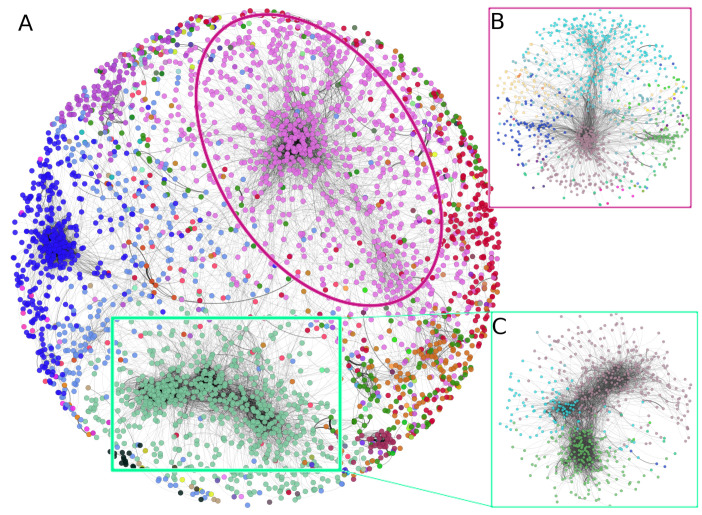
Hierarchical community structure of the PCN. (**A**) Communities at the first level in the PCN. (**B**) Subcommunities of community 2 (pink nodes). (**C**) Subcommunities of community 1.

**Figure 4 cancers-14-02957-f004:**
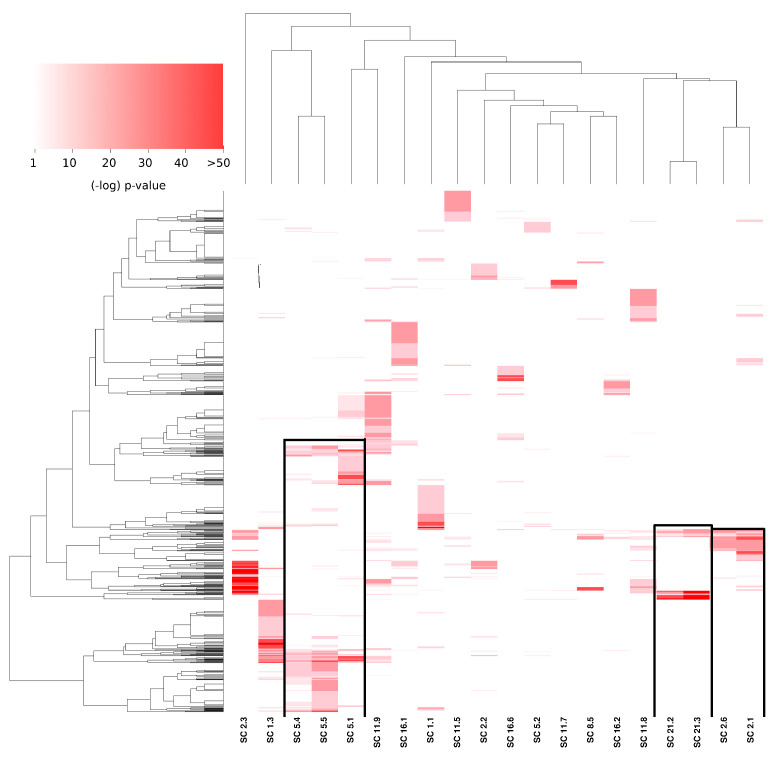
Over-represented communities in the PCN. This heat map shows those biological processes (Y axis) that are statistically associated with specific subcommunities (X axis). The intensity of color is proportional to the significance of the GO category in that subcommunity. Black squares highlight the subcommunities with a similar enrichment pattern. It is worth noticing that communities are mutually exclusive (there are no shared genes between subcommunities).

**Figure 5 cancers-14-02957-f005:**
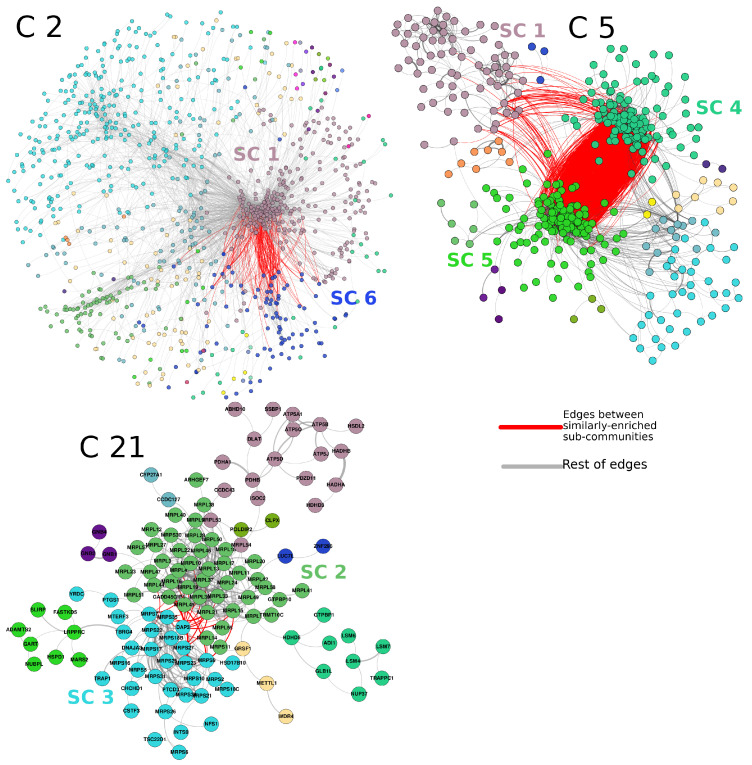
Network structure of three commonly enriched communities and their subcommunities. Each one of the networks shown here is statistically associated (*p*-value < 1 × 10${}^{-5}$) with a common set of biological processes. The color of nodes groups genes of the same subcommunity. Red links represent interactions between genes of the subcommunities with a similar enrichment profile.

**Figure 6 cancers-14-02957-f006:**
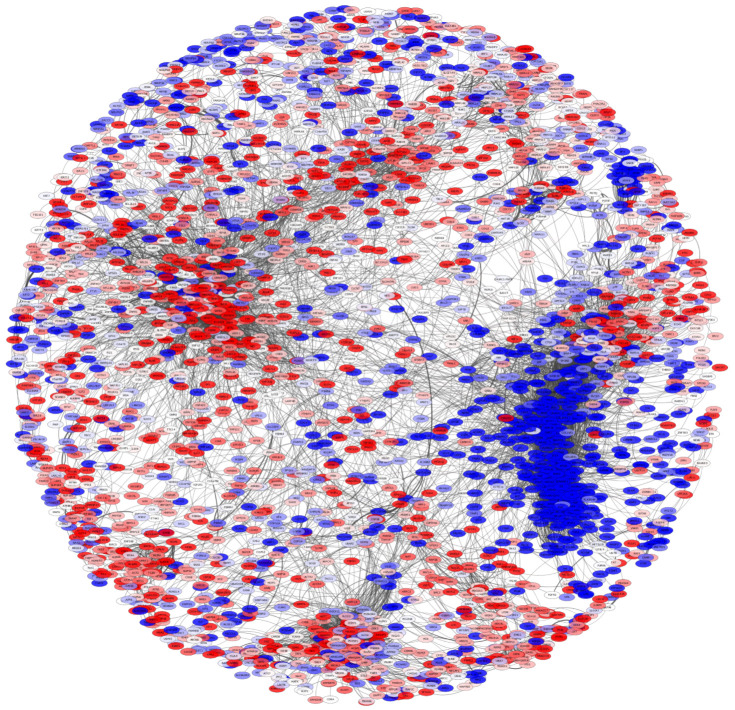
Protein co-expression network. This network shows the pGenes in the giant component of the network. The color of nodes represents the log2 fold-change expression level of genes inferred from the protein expression of tumor samples compared to healthy tissue samples for shared and unshared peptides. Overexpression is depicted in red, while blue nodes show underexpression. Notice the distribution of nodes biased to the differential expression values.

**Figure 7 cancers-14-02957-f007:**
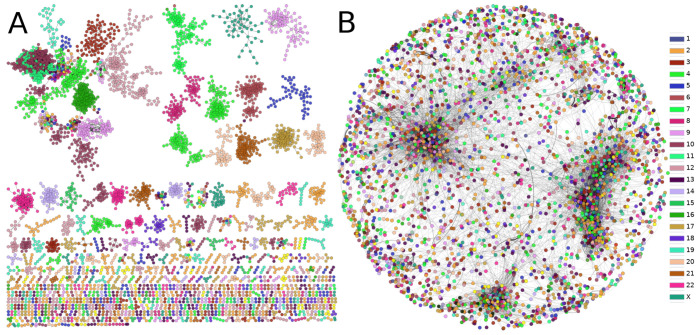
Comparison between (**A**) gene and (**B**) protein co-expression networks. The color of nodes represents the chromosome to which each gene belongs. The loss of inter-chromosome interactions in (**A**) contrasts with the diversity of edges in the PCN. Additionally, the size of the largest component in both cases is largely different.

## Data Availability

The code for this study is available at https://github.com/CSB-IG (accessed on 8 May 2022).

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
