# Peer review of "The Breast Cancer Protein Co-Expression Landscape"

_cancers, 2022, doi:10.3390/cancers14122957_

Round 1

Reviewer 1 Report

The authors present the protein network analysis of breast cancer in an organized manner. They performed the mutual information co-expression analysis with open proteomic breast cancer data. This article provides significant insight into the field of tumor biology. Therefore, I would like to suggest accepting this article for the journal.  

They addressed how proteins of breast cancer cell are related through network analysis with proteomic data. I consider the topic is relevant in the field of tumor biology because they present the characteristics of breast cancer protein network. The conclusions are consistent with the evidence presented and they address the main quesion posed. I would like to suggest the additional table or figure showing conclusive results of protein network of breast cancer.

Reviewer 2 Report

This manuscript by Ruhle et al. describes a workflow of using mutual information between protein levels to infer protein co-expression networks. This proteomics data-mining process aims to capture the co-expression network at the protein level, where some transcriptional co-expression information might be inconstant with actual biological processes. The workflow also aims to rescue the non-linear correlations between variables that may be lost from using traditional dependency measuring methods. The workflow contains four stages: data acquisition and clean-up; protein abundance and pathway differential analysis; using mutual information to construct co-expression network; protein-protein interaction network and enriched community analysis. This multi-layered workflow was applied to breast cancer proteomic data analysis, revealing that at the sub-community level of co-expression, proteins are associated with specific biological processes, and share similar enrichment and interactions. In addition, the authors indicate that protein co-expression is more closely related to the functionality than in the genomic location. My comments below aim to help strengthen the manuscript.

  1. It is somewhat expected that the protein co-expression network will reveal sub-communities’ correspondence to shared biological functions. Did the authors observe any synergies between two different biological functions? What would an example of this inform us with the systems biology of tumor cells?

  1. Since the 4 subtypes of breast cancer were represented in the data acquired, it would be interesting to see any differential co-expression sub-communities can be discovered among these subtypes using the workflow presented. How robust is this workflow when comparing two similar samples? A few sentences in the discussion would be helpful.

  1. Minor issues with spelling and acronyms (e.g. define PCN in the first sentence of the last paragraph in Introduction)

Reviewer 3 Report

Review of Manuscript Cancer-1741528, “The breast cancer protein co-expression landscape”

The manuscript describes the investigation into breast cancer co-expression landscape through the analysis of open proteomic data via mutual information co-expression networks analysis. The author collected mass spectrometry data from 105 breast cancer cases with 3 replicates of tumor tissues and 3 healthy tissues each from Clinical Proteomic Tumor Analysis Consortium. The authors performed 4 main stage for co-expression networks analysis, including 1) Data acquisition, quality control and data curation, 2) Differential protein abundance and pathway analysis, 3) Inference and analysis of protein co-expression networks, and 4) Network modularity analysis and protein interaction networks. The authors observed that that there are distinctive biological processes associated to communities of these networks and how some transcriptional co-expression phenomena is lost at the protein level. This manuscript is easy to read and may provide valuable information to interested readers. Specific comments and suggestions are listed in the following.

1.          The publication, “Yamada, K., Nishimura, T., Wakiya, M., Satoh, E., Fukuda, T., Amaya, K., ... & Ishikawa, T. (2021). Protein co-expression networks were identified from HOT lesions of ER+ HER2–Ki-67high luminal breast carcinomas. Scientific reports, 11(1), 1-13.”, reported the application of protein co-expression networks analysis in breast cancer research. The authors should cite this publication in the revised manuscript. A comparison of the findings and conclusions derived from the publication and the current manuscript under consideration should be provided.

2.          The lines 176stated “The top-10,000 highest MI interactions between proteins in the breast cancer network can be observed in Fig. 3.” The authors did not describe what is the definition of “MI” in the current manuscript. Please clarify it in the revised manuscript.

3.          Method, 2.2 Stage 2: Differential protein abundance and pathway analysis, and Figure 1 right panel (Stage 2), Differential expression analysis. However, there is no data or discussion presented on differential expression analysis. It is suggested that the authors include and describe all relevant data.

4.          Method, line 68, stated “our pipeline starts with data acquisition from the primary databases and preliminary curation and quality control (QC)” and lines 83 to 84 stated “A quality control analysis of the proteogenomic matrix was performed using R.”. However, the manuscript did not include any description to explain the method to control the data quality.

5.          The statement in line 210-212 mentioned the sub-communities with similar enrichment among SC5_1, SC5_4 and SC5_5. However, the SC5_1 and SC5_4 did not show on Figure 4. Please clarify.

6.          The citation style should be consistent. For example, “[8,9]” in line 72 and “(Wickham, 2009)” in line 85.

7.          The citation in line 99, “(Hunter et al. 2012)” and “McKinney et al. 2009” were not listed in the references list. Please clarify.

8.          The citation in line 150, “Gang Su, et.al 2010” was not listed in the references list. Please clarify.

9.          The line 58 and Figure 1 stated, “Luo W et al., 2013” and “Brian McDonagh et al., 2015” were not listed in the references list. Please clarify.

10.      The full name of “PCN” should be provide in line 54.

11.      Typo: Line 68, “dta” should be “data”.

12.      Typo: line58, “Wit” should be “With”.

13.      Typo: line 210 “SSC5_4” should be “SC5_4

Round 2

Reviewer 3 Report

The manuscript is very much improved. The authors have made necessary modifications in the revised manuscript to properly address the comments. Therefore, it is suggested that the revised manuscript be accepted for publication without further revision.